# E^2^LNet: An Efficient and Effective Lightweight Network for Panoramic Depth Estimation

**DOI:** 10.3390/s23229218

**Published:** 2023-11-16

**Authors:** Jiayue Xu, Jianping Zhao, Hua Li, Cheng Han, Chao Xu

**Affiliations:** School of Computer Science and Technology, Changchun University of Science and Technology, Changchun 130022, China; 2021200124@mails.cust.edu.cn (J.X.); lihua@cust.edu.cn (H.L.); hancheng@cust.edu.cn (C.H.); 2018200084@mails.cust.edu.cn (C.X.)

**Keywords:** panoramic depth estimation, dilated convolution, global average pooling, pixel-wise attention

## Abstract

Monocular panoramic depth estimation has various applications in robotics and autonomous driving due to its ability to perceive the entire field of view. However, panoramic depth estimation faces two significant challenges: global context capturing and distortion awareness. In this paper, we propose a new framework for panoramic depth estimation that can simultaneously address panoramic distortion and extract global context information, thereby improving the performance of panoramic depth estimation. Specifically, we introduce an attention mechanism into the multi-scale dilated convolution and adaptively adjust the receptive field size between different spatial positions, designing the adaptive attention dilated convolution module, which effectively perceives distortion. At the same time, we design the global scene understanding module to integrate global context information into the feature maps generated using the feature extractor. Finally, we trained and evaluated our model on three benchmark datasets which contains the virtual and real-world RGB-D panorama datasets. The experimental results show that the proposed method achieves competitive performance, comparable to existing techniques in both quantitative and qualitative evaluations. Furthermore, our method has fewer parameters and more flexibility, making it a scalable solution in mobile AR.

## 1. Introduction

Depth estimation is a crucial task in the field of computer vision that involves predicting the depth information of pixel-wise in an image. With the increasing popularity of 360° cameras, monocular panoramic depth estimation has become widely utilized in various fields such as robot navigation, autonomous driving, virtual/augmented reality [1,2], and more, due to its capability of providing a comprehensive view of the scene. As a result, it serves as the foundation for scene understanding and object detection in these applications.

With the widespread application of deep learning in computer vision, convolutional neural networks (CNNs) have become a primary method for processing image depth estimation. However, early studies [3,4] have shown that using depth neural networks trained on perspective images directly on panorama images results in inferior performance. This is because panorama images experience geometric distortion, which makes panoramic depth estimation a challenging research topic. In particular, omnidirectional images contain 360° FoV environmental information, making them more complex than near-field-of-view (NFoV) images. Panorama images are typically represented by expanding a sphere into a 360° × 180° equirectangular projection (ERP). Due to the varying spatial sampling rate with latitude, distortion increases from the equator to the polar regions. Therefore, traditional CNNs may struggle to perform accurate depth estimation in distorted areas.

In recent years, researchers have made attempts to tackle distortion by creating convolution kernels that are specifically designed for this purpose [3,4,5,6]. However, the fixed sampling positions of CNNs can restrict their ability to extract features effectively. Additionally, some studies [7,8,9,10] have utilized projection–fusion to decrease distortion, but this method often leads to overlapping projection regions between different projections, resulting in unnecessary computational overheads.

Another challenge in monocular panoramic depth estimation is the lack of global consistency in convolutional neural networks. Although current CNN-based methods aim to address distortion, their fixed receptive field limits their ability to perceive global information in panorama images. One approach to preserving global information is SliceNet [11], which uses a slice representation that directly utilizes the characteristics of the equirectangular projection of indoor scenes, without the need for distortion-aware convolution and transformation. However, SliceNet sacrifices detailed information when reconstructing the depth map. Other methods [12,13,14,15] use multi-scale dilated convolutions to gather multi-scale global information, while ACDNet [16] enhances the receptive field in equirectangular projection using adaptively combined dilated convolution and incorporating an adaptive channel-wise fusion module to capture contextual information. However, embedding multi-scale dilated convolutions in the encoder ResNet50 may diminish its feature extraction ability.

To overcome the above-mentioned limitations, we propose an efficient and effective lightweight network for monocular panoramic depth estimation, called $E^{2}L$Net. Our framework addresses both the distortion problem and the lack of global contextual information, leading to better performance. $E^{2}L$ Net uses the U-Net [17] architecture and incorporates an adaptive attention dilated convolution module at the skip connections.This module introduces an adaptive pixel-wise attention that enables the network to learn dependencies between different positions and adjust the receptive field size dynamically. Additionally, we designed a global scene understanding module that captures global contextual information. Unlike [16], our module is added at the bottleneck of the network and does not affect the feature extraction capability of the encoder, making it easier to replace existing excellent backbone networks.

In summary, our contributions are as follows:We propose an efficient framework, $E^{2}L$Net, for monocular panoramic depth estimation that can simultaneously address distortion and extract global contextual information;We design the adaptive attention dilated convolution module to be added at the skip connections, enabling distortion perception at different scales without disrupting the internal structure of the encoder and preserving its feature extraction ability;We construct a global scene understanding module by utilizing multi-scale dilated convolutions, which effectively capture comprehensive global information;We conduct panoramic depth evaluation experiments on both virtual and real-world RGB-D panorama datasets, and our proposed model achieves results comparable to existing methods.

The rest of this paper is organized as follows. Section 2 reviews the related work on monocular panoramic depth estimation as well as attention mechanisms and dilation convolutions. Section 3 describes our proposed $E^{2}L$Net model approach. Section 4 discusses the experimental results, including quantitative, qualitative, and complexity analysis. Finally, Section 5 provides a summary of the paper.

## 2. Related Work

In this section, we reviewed the research overview of panoramic depth estimation. We also provided a brief introduction to the application of attention mechanisms and dilated convolutions in CNNs.

### 2.1. Monocular Panoramic Depth Estimation

Depth estimation from perspective images is a well-known problem, and significant progress has been made in the past decade using deep-learning-based monocular perspective depth estimation [18]. As perspective images have a limited field-of-view, panorama images contain more global spatial information, and using panorama images for depth estimation can lead to better 3D scene reconstruction and understanding. With the increasing popularity of 360° cameras, monocular panoramic depth estimation has been widely applied in VR, AR, autonomous driving, and other fields. However, directly applying the deep learning methods used for perspective images to panoramic images with distortion issues is not feasible [3].

In recent years, several methods have been proposed to address distortion issues in monocular panoramic depth estimation. One approach is to use special convolution operations [3,4,5,6]. Tateno et al. [4] introduced a deformable convolution filtering method for dense prediction tasks, which allows training a CNN with regular convolution and perspective images, and then transferring the weights to another network with the same architecture that can perform deformable convolution to solve distortions in ERP images. However, their approach neglects the panoramic wide field-of-view, leading to prominent artifacts. Cheng et al. [19] proposed an omnidirectional depth expansion convolutional neural network, which embeds a spherical feature transformation layer at the end of the feature encoding layer to resample the neighborhood of each pixel in the omnidirectional coordinate to the projection coordinate, reducing the difficulty of feature learning. They also appended a deformable convolution spatial propagation network at the end of the feature decoding layer, significantly improving visual quality. Another method to reduce distortion is through projection–fusion [7,8,9,10]. Wang et al. [7] proposed a dual projection–fusion method called BiFuse, which extracts and fuses features through two neural network branches for equirectangular projection and cube projection, respectively. The equirectangular branch provides a complete field-of-view, while the cube mapping avoids local distortion. A spherical padding is used to alleviate the discontinuity at the boundary of the cube projection. BiFuse++ [9] combines dual projection–fusion with self-supervised learning of scenes to further improve BiFuse’s performance. To reduce the complexity of the BiFuse model, UniFuse [8] only feeds the cube mapping features into the equirectangular features at the decoding stage.

To extract global context information of panoramas, some transformer-based methods are used to estimate globally consistent depth maps [10,20,21]. PanoFormer [20] reshapes the self-attention module with the learnable token flow to adapt panoramas. GLPanoDepth [10] utilizes a cubemap vision transformer (CViT) and a CNN to extract global and local features, respectively. Different features are progressive, combined by a gated fusion module. PCFormer [21] adopts channel attention and spatial attention to enhance global and local features. These methods utilize attention mechanisms to selectively focus on important regions and learn more effective global representations.

Additionally, several panoramic depth estimation methods have been proposed that use different strategies. For example, Zioulis et al. [22] utilized a depth-image-based rendering (DIBR) technique to merge panorama images with their corresponding depth maps to create new synthesized views that simulate translational (vertical or horizontal) camera motion. They also proposed a self-supervised scheme that minimizes depth-based photometric error instead of depth itself. Similarly, Zeng et al. [23] used coarse depth and semantic predictions to perform layout depth prediction. They then used the estimated layout depth map to recover the 3D layout and refine the depth estimation. Jin et al. [24] leveraged the correlation between depth and geometry of 360° indoor images. They represented the geometry structure of indoor scenes as a set of corners, boundaries, and planes, and used the geometry structure and regularizer for depth estimation. Furthermore, Zhou et al. [25] proposed a robust panoramic depth estimation network called PADENet. They combined the fundamental envelope loss and window-based loss to improve the loss of PADENet.

Due to the intricate nature of indoor scene structures, the existing methods for panoramic depth estimation often rely on constructing complex and computationally heavy networks. In contrast, our proposed approach introduces an efficient and effective lightweight network for panoramic depth estimation.

### 2.2. Dilated Convolution

Dilated convolution is a widely used technique in convolutional neural networks to increase the receptive field without adding more parameters. Dilated convolution first appeared in semantic segmentation [26] and significantly improved the performance of segmentation. It has been increasingly applied to depth estimation tasks [15,27,28,29,30] in recent years, such as in DORN [27], which is a deep depth-ordered regression network. DORN proposed a multi-scale feature learner that utilizes three dilated convolution layers with large kernel size, a 1×1 convolution layer, and a full-image encoder to capture global context information. Tian et al. [28] designed the continues layers with prime dilation rate, allowing features to propagate from input to output without skipping any pixels while maintaining the original resolution. Another example is the attention-based contextual aggregation network designed by Chen et al. [12]. This network uses dilated convolutions and an attention model to capture contextual information for each pixel. Although the multi-branch network structure can reduce the speed of network operation, it improves the accuracy and generalization of depth estimation. Recently, some depth estimation tasks [16,31,32] have aggregated multi-scale contextual information by introducing ASPP [33].

Dilated convolution has been applied to panoramic depth estimation tasks, for example, OmniDepth [3] is a network that utilizes this technique. It is a fully convolutional encoder–decoder network that increases the receptive field and adapts to the distortions of ERP images with different sizes of rectangular convolutions. However, OmniDepth’s limitation lies in its use of only two down-sampling operations, which leads to a restricted receptive field. As a result, it fails to extract a sufficient amount of contextual information. To address this, Zhuang et al. [16] proposed a new framework that combines dilated convolution with different dilation rates to increase the receptive field while reducing distortion. In addition to depth estimation tasks, dilated convolution has also been applied in other panoramic vision tasks [34,35].

Overall, the use of dilated convolution has shown promising results in various panoramic depth estimation vision tasks. Furthermore, the combination of different dilation rates and contextual information can further improve performance in real-world scenarios [36,37,38]. Therefore, inspired by [27], we propose a new scene understanding module that utilizes global pooling and multi-scale dilated convolution operations to capture global contextual information in panoramas.

### 2.3. Pixel-Attention Model

The attention mechanism [39] is a crucial concept in computer vision and has been widely used to process computer vision tasks. In recent years, several methods have been proposed to incorporate attention mechanisms into computer vision models. For instance, the SE attention mechanism [40] (squeeze-and-excitation network), utilizes an additional neural network to automatically learn the importance of each channel in the feature map. It assigns a weight value to each feature based on its importance, allowing the neural network to focus on the most relevant feature channels. The CBAM attention mechanism [41] (convolutional block attention module) combines channel attention and spatial attention to learn the importance weights for each channel and capture the importance of each position in the feature map. In panorama image processing, many related works [12,14,15,42,43,44] have utilized attention mechanisms. Chen et al. [12] utilized self-attention mechanisms to acquire pixel-level contextual information and combined it with image-level contextual information for monocular depth estimation. Zhao et al. [42] introduced a learnable attention mechanism that learns binary decision variables through the Gumbel–Max trick in a differentiable training framework. Jiao et al. [14] novel attention mechanism which allows to robustly fuse features derived from multiple scales as well as to integrate structured information. Chen et al. [45] proposed generating attention weights for each pixel in the semantic segmentation task. Pixel-level attention is a soft attention model that can help the model focus on important pixels in the image. Inspired by [16], we introduce pixel-wise attention to action on the features extracted by multi-scale extended convolution to better perceive the distortion of panorama images. Our experiments show that the pixel-wise attention mechanism allows the model to selectively focus on certain pixels, which is particularly useful for distortion perception of panoramic images.

## 3. Proposed Algorithm

### 3.1. Network Architecture

The central concept of our proposed approach is to tackle both the distortion in panorama images and the extraction of global contextual information in a single, unified method. Figure 1 provides an overview of the proposed $E^{2}L$Net architecture, which is based on an encoder–decoder framework with skip connections. The $E^{2}L$Net main introduce two modules: (1) An adaptive attentional dilated convolution (AADC) module with distortion awareness. By introducing a pixel-wise attention mechanism that enables adaptive adjustment of receptive fields for channels and spatial positions, the ability to perceive panoramic distortions is enhanced. This allows for a more effective handling of distortions in panoramic images, improving the accuracy of depth estimation. (2) A global scene understanding module (GSUM) for global context extraction. Given the expansive field of view in panorama images, global information plays a crucial role in achieving accurate depth estimation. The incorporation of global context information into the feature extraction process aids the model in gaining a deeper understanding of the overall scene context and structure, bolstering the robustness of panoramic depth estimation. The integration of these two modules can greatly improve the performance of panoramic depth estimation.

In the encoder part of the model, ResNet34 [46] is utilized for feature extraction. Compared to other neural networks, ResNet34 has a shallower network structure and fewer parameters. Despite this, it has comparable accuracy to deeper networks in many tasks.Given the input panoramic color image, the encoder uses the residual structure of ResNet34 to extract multi-scale feature maps. We collect multi-scale feature maps obtained from the residual blocks of ResNet34 with are denoted as $F_{1}$, $F_{2}$, $F_{3}$ and $F_{4}$.The spatial sizes of the multi-scale feature maps are $\frac{H}{4}\times \frac{W}{4}\times C$, $\frac{H}{8}\times \frac{W}{8}\times 2C$, $\frac{H}{16}\times \frac{W}{16}\times 4C$, $\frac{H}{32}\times \frac{W}{32}\times 8C$, where the value of *C* is 64.

In the decoder part of our proposed $E^{2}L$Net, we utilize the PixelShuffler [47] to progressively up-sample the feature maps and aggregate the feature maps generated by the encoder at different scales. This allows us to effectively capture and integrate multi-scale features while maintaining computational efficiency. To capture global information while considering computational costs, we introduce a GSUM module to the bottleneck. The GSUM module allows us to extract global context information while minimizing the impact on the overall model complexity. We will provide a more detailed explanation of this module in Section 3.3. To further perceive multi-scale distortion, we perform an adaptive attentional dilated convolution operation on the multi-scale features obtained by the encoder. The AADC operation helps us better perceive and handle distortion in panorama images. Through skip connections, the distortion-aware feature maps obtained by the AADC operation are concatenated with the up-sampled feature maps in the decoder with the same number of channels. This allows us to effectively integrate multi-scale features with distortion-aware information. We will provide a detailed explanation of the AADC module in Section 3.2.

### 3.2. Adaptively Attention Dilated Convolution

The idea for the AADC module, which utilizes the pixel-wise attention, inspired by ACDNet [16]. As depicted in Figure 2a, ACDNet incorporates an adaptive dilated convolution module within the encoder that replaces the standard convolution with a residual structure. After passing the input features through multi-scale dilated convolutions, four feature maps are obtained. These feature maps are then concatenated to form feature map $F_{i}$. Following this, global average pooling and fully connected operations are applied, and Softmax is used to generate attention weights $w_{i}$. These attention weights are multiplied with the input features to obtain the fused feature $F_{i}^{\prime }$. We designed a module as shown in Figure 2b. We noticed that the attention output dimension for channel attention is 1×1×C, for spatial attention is H×W×1, and for pixel attention is H×W×C. Despite the increased number of parameters, we can obtain both channel and spatial attention simultaneously. Different from the adaptive channel-wise fusion (ACF) method in ACDNet, we introduced the adaptive pixel-wise fusion (APF) module to aggregate the feature maps and obtain different attention regions in the receptive field along the channel and spatial positions. Additionally, we placed the AADC module between the encoder and decoder, without compromising the encoder’s feature extraction ability.

To more comprehensively extract distortion-aware feature maps, the AADC module we designed combines pixel-wise attention and multi-scale dilated convolution operations, as shown in Figure 2b, and the feature fusion process is as follows: (1)$$F_{i}^{k}=AdaptiveDilatedConv\left(F_{i}\right),k=1,2,3,4$$
(2)$$w_{i}=PixelwiseAttention\left(F_{i}\right)$$
(3)$$F_{i}^{\prime }=w_{i}⊙Addtion\left(F_{i}^{k}\right)$$
where ⨀ is element-wise product and models the cross-relationships between features from two branches. The AADC module contains two branches. Firstly, given the input feature $F_{i}$, a set of dilated convolutions with different dilation rates (kernel size 3×3, dilation rates 1×1, 1×2, 1×4, 2×1) are used in parallel to extract features from the input feature, and then combined by addition. Then, the APF module integrates pixel features with the feature map. Specifically, the pixel features are first normalized to the range of 0–1 by a 1×1 convolution and the Sigmoid function to obtain pixel-wise attention weights $w_{i}$. The feature map $F_{i}^{k}$ after adaptive dilated convolutions is then multiplied with the pixel-wise attention weights $w_{i}$ to obtain the final fusion feature $F_{i}^{\prime }$ with distortion awareness.

### 3.3. Globle Scene Understandng

Our GSUM module is inspired by [27] and is designed to efficiently capture global context information with minimal computational resources. It is composed of three parallel components, as illustrated in Figure 3.

In the GAP module, we efficiently extract global information with fewer parameters. To address the issue of local ambiguity in depth estimation [48], we perform global average pooling on the input feature map $F_{4}$ to obtain a feature vector. We then use a 1×1 convolutional layer to allow channel-wise information interaction, and finally, we use replication operations to obtain the full feature map from the feature vector. We use a 1×1 convolutional layer to facilitate channel-wise information interaction. We utilize a multi-scale adaptive dilated convolution module to extract features by increasing the receptive field at different scales. We use four dilated convolutions with different dilation rates of 1×1, 1×2, 1×4 and 2×1, and a kernel size of 3×3 for feature extraction. This is because, in panorama images, horizontal distortion near the extreme point is greater than the vertical distortion. Through the above operations, each position in the input feature map can understand the image in the same way. After connecting the feature maps generated by the three above components, we apply a 1×1 convolution operation to obtain the output feature map ${\hat{F}}_{4}$, which enables a comprehensive understanding of the input image.

### 3.4. Training Loss

Following previous works on panoramic depth estimation [7,8], we utilize the robust BerHu loss [49] as the objective function to supervise the training process of the network, which is formulated as follows.
(4)$$L\left(d,g\right)=\left\{\begin{array}{cc} |d-g|, & |d-g|\leq c\\ \frac{{\left(d-g\right)}^{2}+c^{2}}{2c}, & |d-g|>c \end{array}\right.$$
where *d* is the predicted depth and *g* is the ground truth depth. The threshold *c* is set to 20% of the maximal absolute error of the current batch in every gradient descent step as in [49], that is
(5)$$c=\frac{1}{5}\max_{i}\left(\left|d_{i}-g_{i}\right|\right)$$
where *i* indexes all pixels over each image in a batch.

## 4. Experiments

In this section, we first introduce our experimental setting, including the datasets, training details and evaluation metrics. Second, we provide the qualitative and quantitative comparisons of the proposed method with state-of-the-art approaches. Then, we conduct the ablation experiments to validate the effectiveness of our network structure. Finally, we test the complexity of our model.

### 4.1. Experimental Settings

**Datasets:** Our experiments are carried out on three real-world and synthetic datasets, namely Matterport3D [50], Stanford2D3D [51], and PanoSUNCG [52]. Both Matterport3D and Stanford2D3D are captured by RGB-D cameras in real-world scenes, and they contain 10,800 and over 2000 panoramas and corresponding depth maps, respectively. Similar to many recent methods [7,8], we use their official splitting and resize the resolution of RGB images and depth maps into 512 × 1024. Because Matterport3D is missing the top and bottom depth values and Stanford2D3D is missing the ceiling and floor pixels in the RGB images, which results in inaccurate depth predictions for these regions, we utilize binary masks to filter out the depth values predicted for these regions. PanoSUNCG is rendered with synthetic scenes, and it contains 103 scenes and over 25,000 RGB-D panorama images of 512 × 1024 resolution. Follow the official training and testing splits, 80 scenes are used for training and 23 scenes for testing. As with the works [9], all RGB-D panorama images are resized to 256 × 512 resolution during training and testing. Besides, these three datasets are publicly available on their official websites.

**Training details:** We use the PyTorch framework [53] to implement our network. All experiments are conducted on an Intel Xeon(R) CPU processor, 128 GB RAM, and an NVIDIA GeForce RTX 3090 GPU with 24 GB memory. The ResNet34 encoder of the network is initialized with the weights pre-trained on ImageNet [54], and the remaining layers are initialized uniformly. The Adam optimizer [55] is used in our experiments with default parameters $\beta_{1}$ = 0.9, $\beta_{2}$ = 0.999, the initial learning rate is set to 0.0001, and the batch size is set to 8. The total training epochs are set to 100 for real-world datasets and 30 for synthetic datasets. Since the real-world dataset is smaller than the synthetic datasets, more training epochs are required to produce a desirable result. To decrease the possibility of network over-fitting, we use several basic data augmentations during training, such as random color adjustment, left–right flip, and yaw rotation.

**Evaluation metrics:** We use the standard metrics for evaluation as in previous works [7] and adopt the following quantitative evaluation metrics in our experiments:Mean relative error (MRE):
(6)$$\begin{array}{c} \frac{1}{N}\sum \frac{\left|d-g\right|}{g} \end{array}$$Mean absolute error (MAE):
(7)$$\begin{array}{c} \frac{1}{N}\sum \left|d-g\right| \end{array}$$Root mean square error (RMSE):
(8)$$\begin{array}{c} \sqrt{\frac{1}{N}\sum {\left(d-g\right)}^{2}} \end{array}$$Root mean squared log error (RMSE log):
(9)$$\begin{array}{c} \sqrt{\frac{1}{N}\sum {\left(\log\;d-\log\;g\right)}^{2}} \end{array}$$Accuracy with threshold *t*:
(10)$$\begin{array}{c} max\left(\frac{d}{g},\frac{g}{d}\right)=\delta <t\left(t\in \left[1.25,1.25^{2},1.25^{3}\right]\right) \end{array}$$
where *d* is the predicted depth, *g* is the ground truth depth, and *N* is the number of valid depth value in the ground truth depth. The lower the error metrics are, the better; the higher the accuracy metrics are, the better.

### 4.2. Comparison Results

In this section, we conduct a comparison between our method and four existing approaches: Bifuse [7], UniFuse [8], BiFuse++ [9], and ACDNet [16]. We present the quantitative results in Table 1, while Figure 4, Figure 5 and Figure 6 showcase the qualitative results for the three benchmark datasets.

**Quantitative results:** In this section, we compare our method with Bifuse [7], UniFuse [8], BiFuse++ [9], and ACDNet [16] on the three mentioned datasets. The quantitative results are presented in Table 1. Across all three datasets, our method consistently outperforms the other approaches in the majority of the numerical metrics. Although our method shows slightly lower numerical values in metrics such as RMSE, $\delta <1.25^{2}$, and $\delta <1.25^{3}$ compared to the state-of-the-art approaches; this can be attributed to the lower parameter count in our model. However, our method consistently outperforms the state-of-the-art algorithms in the most stringent accuracy metric, δ<1.25. Specifically, on the real-world datasets, Matterport3D and Stanford2D3D, our method achieves improvements of 0.68% and 0.22% respectively over the state-of-the-art approaches. Unfortunately, due to the unavailability of the synthetic dataset PanoSUNCG for comparison with ACDNet, we compared our method with other advanced algorithms on that dataset. Remarkably, our method surpasses the state-of-the-art BiFuse++ by 1.31% in the strictest accuracy metric, δ<1.25, while also reducing the RMSE error by 0.0181 compared to the leading algorithm. These evaluation results demonstrate that our method achieves enhanced strict accuracy while simultaneously reducing the parameter count. We have also presented the convergence curves for training loss and δ<1.25 accuracy, RMSE, and MAE on the validation set, as shown in Figure 7. The Stanford2D3D dataset, with its rendered depth information, offers higher precision and is relatively easier to learn from. However, its limited size, consisting of just over 2000 images, results in lower evaluation accuracy. Conversely, the PanoSUNCG dataset is a virtual dataset that is more easily learnable and provides higher evaluation accuracy. Figure 7a,b illustrates the rapid convergence of our model to a high precision level, demonstrating that there is no over-fitting in terms of accuracy and error on the validation set.

**Qualitative results:** In Figure 4 and Figure 5, we present a qualitative comparison of our method with BiFuse++ and ACDNet on real-world datasets, including Matterport3D and Stanford2D3D. Upon observation, we find that BiFuse++ tends to introduce artifacts or lose certain objects, while ACDNet generates depth maps with a certain degree of blurriness. In contrast, our method exhibits a superior ability to accurately reconstruct these objects, thanks to its advanced distortion awareness and capability to capture global information. In Figure 6, we conduct a qualitative comparison of our method with UniFuse and BiFuse++ on the synthetic dataset, PanoSUNCG. Our findings reveal that UniFuse often produces erroneous depth estimates, as evident from the incorrect estimation of carpet depth in the first row. On the other hand, BiFuse++ tends to result in the loss of depth details for small objects. Comparatively, our method produces depth estimates that closely align with the ground truth. Additionally, We represent the residual map as the absolute difference between the predicted depth and the ground truth. To enhance the visual clarity, we have applied an inversion to the residual map. Following the previous approach, it is important to note that we did not perform scale calibration during the measurements. Consequently, there may be a slight discrepancy between our estimated scale and the true scale. The error map further illustrates the offset between the estimated depth map and the ground truth depth map across the entire depth range.

### 4.3. Ablation Study

We conducted a series of ablation experiments on the Matterport3D dataset to assess the effectiveness of our proposed model. Table 2 presents the results of these experiments, which involved different configurations of network modules. Specifically, we compared the performance of four models across various metrics, with the baseline model representing a UNet architecture using only ResNet34 as the encoder.

Firstly, we examined the efficacy of the global scene understanding module. This module employed global average pooling to capture pixel-level features from the feature maps, enabling better extraction of global information.

Secondly, we evaluated the effectiveness of the adaptive attentional dilated convolution module. Our proposed AADC module outperformed ACDNet [16], which replaces the residual structure of conventional convolutions with adaptive combination dilated convolutions. The AADC module achieved higher accuracy and lower error rates. Among all the models tested, the one incorporating both the GSUM and AADC modules demonstrated the best performance in terms of the strictest accuracy metric. This finding further emphasizes the significance of GSUM and AADC in enhancing the model’s performance.

### 4.4. Complexity Comparison

In Table 3, we assessed the complexity of our model compared to previous panoramic depth estimation models. The approach using BiFuse [8] with dual projection–fusion slightly increases the parameter count, consuming 4003 M of GPU memory. Moreover, the dual projection–fusion requires multiple projection transformations, leading to longer inference times for depth map estimation. By simplifying and improving the decoder [7,9], the model’s complexity reduced significantly, resulting in a substantial increase in the inference frame rate. ACDNet [16] incorporates the adaptively combined dilated convolution into ResNet50, resulting in 87.0 M parameters. Our model achieves similar performance to ACDNet but with only half the number of parameters and a more than three-fold increase in frame rate. These findings indicate that our model is suitable for deployment in mobile applications. Currently, the overall accuracy of scene depth calculation is approaching or surpassing 90%, greatly facilitating and encouraging further research in the field of scene understanding.

## 5. Conclusions

In this paper, we present a highly efficient and compact network for monocular panoramic depth estimation. Our network employs multi-scale dilated convolutions to enhance the receptive field and reduce distortion, while effectively capturing global features through pixel-wise attention. We have also devised a global scene understanding module that efficiently acquires global information using global average pooling operations, requiring minimal computational resources. Extensive experiments validate the effectiveness of our proposed model. The final model achieves comparable performance to state-of-the-art methods on three different real and synthetic datasets. Additionally, we demonstrate that our model exhibits lower complexity compared to alternative approaches. Moreover, our designed modules can be easily integrated into the bottleneck of the network, enabling convenient replacement of other high-quality backbone networks.

## Figures and Tables

**Figure 1 sensors-23-09218-f001:**
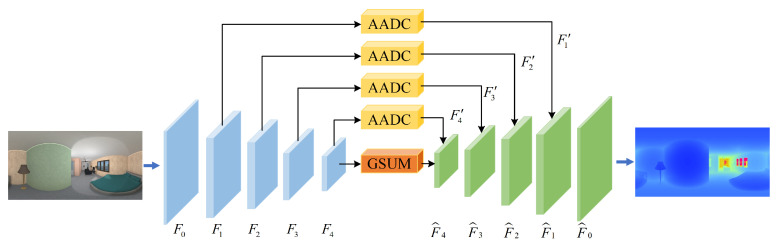
The network architecture of our $E^{2}L$Net. Our $E^{2}L$Net architecture consists of an encoder–decoder. The ResNet network is utilized to extract multi-scale local features, and a GSUM module is employed to extract global information. Adaptive attentional dilated convolutions are used at the skip connections to reduce image distortion. The feature maps processed by GSUM and AADC are concatenated and passed through the decoder to generate depth features at different scales, which are then used for depth estimation.

**Figure 2 sensors-23-09218-f002:**
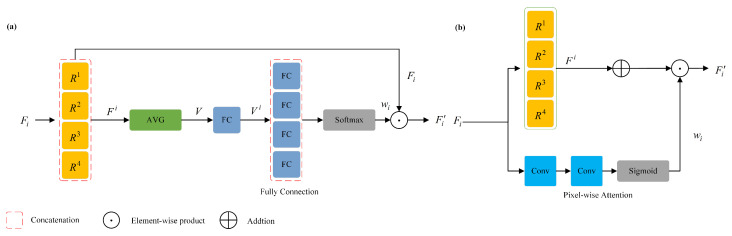
(**a**) The dilated convolution layer of ACDNet. (**b**) The proposed adaptive attention dilated convolution layer in this paper. $R^{n}$ in the figure means the n-th choice of the four dilation rate settings.

**Figure 3 sensors-23-09218-f003:**
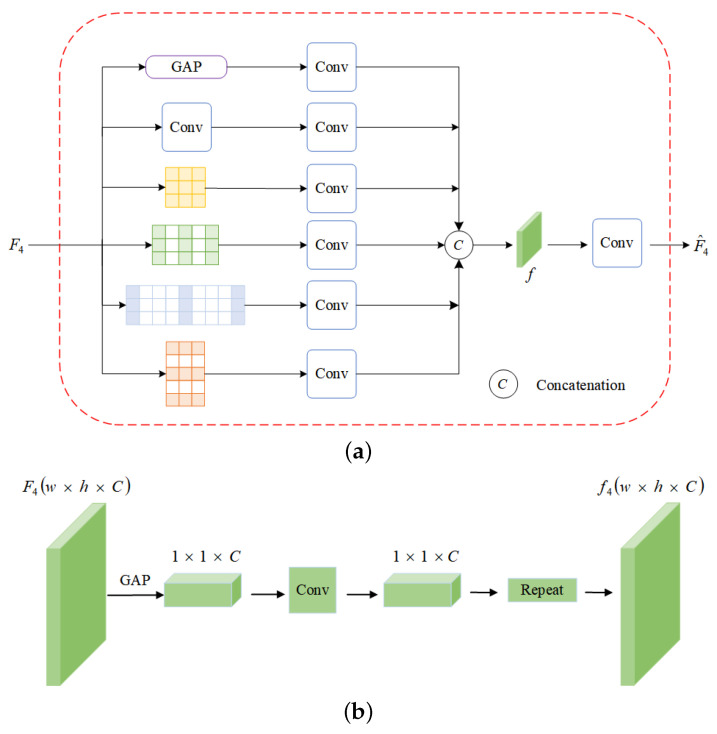
(**a**) The GSUM module structure; (**b**) the global average pooling structure.

**Figure 4 sensors-23-09218-f004:**
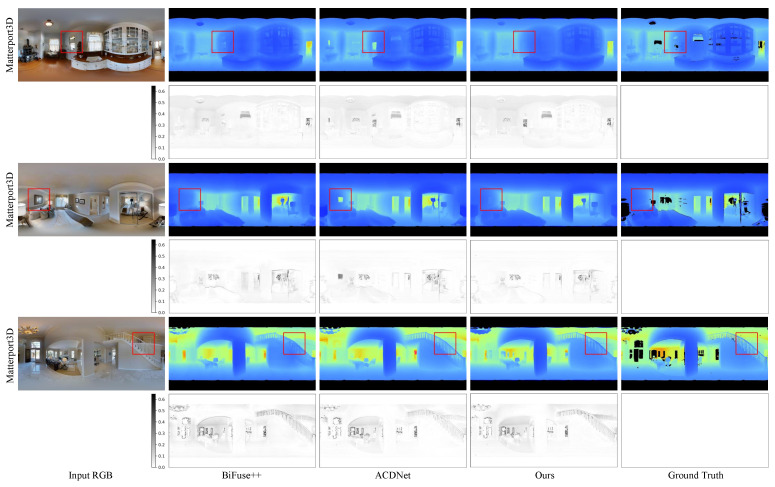
Qualitative comparison with the state-of-the-art methods on Matterport3D dataset. The first column is the input RGB image, the second one is the depth estimated by BiFuse++ [9], the third one is the depth estimated by ACDNet [16], the fourth one is the depth estimated by our method, and the last one is the ground truth depth map. Dark pixels are missing depth in the ground truth depth maps. The residual map is the error map between the predicted depth map and the ground truth depth map. Zoom in for best view.

**Figure 5 sensors-23-09218-f005:**
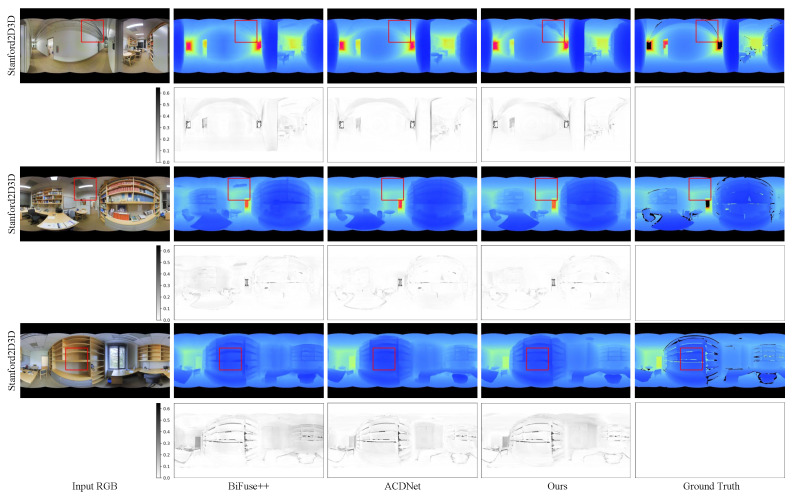
Qualitative comparison with the state-of-the-art methods on Stanford2D3D dataset. It can be observed that BiFuse++ tends to introduce artifacts or lose certain objects, while ACDNet generates depth maps with a certain degree of blurriness. Zoom in for best view.

**Figure 6 sensors-23-09218-f006:**
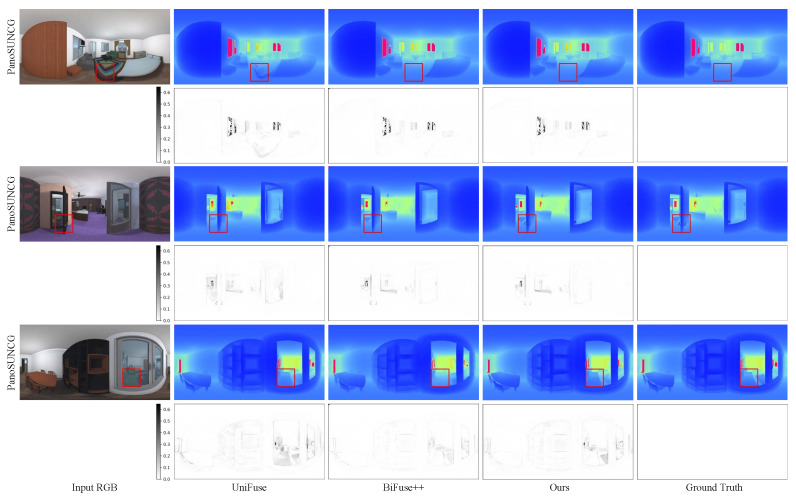
Qualitative comparison with the state-of-the-art methods on PanoSUNCG dataset. The first column is the input RGB image, the second one is the depth estimated by UniFuse [8], the third one is the depth estimated by BiFuse++ [9], the fourth one is the depth estimated by our method and the last one is the ground truth depth map. Pink pixels indicate larger depth values. The residual map is the error map between the predicted depth map and the ground truth depth map. It can be observed that UniFuse often produces incorrect depth estimates, while BiFuse++ leads to the loss of depth details for small objects. Zoom in for best view.

**Figure 7 sensors-23-09218-f007:**
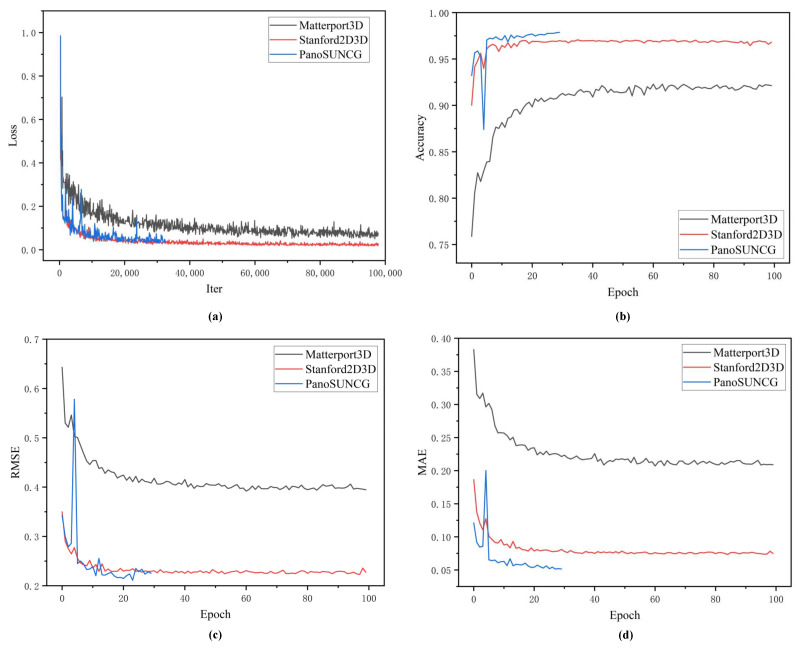
Convergence performance of model. (**a**) The convergence curve for training loss. (**b**) shows the convergence curve for δ<1.25 accuracy on the validation set. (**c**) The convergence curve for validation RMSE. (**d**) The convergence curve for validation MAE.

**Table 1 sensors-23-09218-t001:** Quantitative comparison with the state-of-the-art methods on three benchmark datasets. ↓ indicates that lower is better; ↑ indicates that higher is better. The best results are in **bold**.

Datasets	Methods	Error Metric ↓				Accuracy Metric ↑		
		**MRE**	**MAE**	**RMSE**	**RMSE log**	δ<1.25	$\delta <1.25^{2}$	$\delta <1.25^{3}$
Matterport3D	BiFuse [7]	0.2048	0.3470	0.6259	0.1134	0.8452	0.9319	0.9632
	UniFuse [8]	-	0.2814	0.4941	0.0701	0.8897	0.9623	0.9831
	BiFuse++ [9]	0.1424	0.2842	0.5190	0.0862	0.8790	0.9517	0.9772
	ACDNet [16]	-	0.2670	**0.4629**	**0.0646**	0.9000	**0.9678**	**0.9876**
	Ours	**0.0958**	**0.2610**	0.4661	0.0649	**0.9068**	0.9652	0.9856
Stanford2D3D	BiFuse [7]	0.1209	0.2343	0.4142	0.0787	0.8660	0.9580	0.9860
	UniFuse [8]	-	0.2082	0.3691	0.0721	0.8711	0.9664	0.9882
	BiFuse++ [9]	0.1117	0.2173	0.3720	0.0727	0.8783	0.9649	0.9884
	ACDNet [16]	-	0.1870	**0.3410**	**0.0664**	0.8872	**0.9704**	**0.9895**
	Ours	**0.1094**	**0.1815**	0.3420	0.0673	**0.8890**	0.9614	0.9866
PanoSUNCG	BiFuse [7]	0.0592	0.0789	0.2596	0.0443	0.9590	0.9823	0.9907
	UniFuse [8]	-	0.0765	0.2802	0.0416	0.9655	0.9846	0.9912
	BiFuse++ [9]	0.0524	0.0688	0.2477	0.0414	0.9630	0.9835	0.9911
	Ours	**0.0343**	**0.0484**	**0.1871**	**0.0318**	**0.9761**	**0.9892**	**0.9941**

**Table 2 sensors-23-09218-t002:** Ablation studies about different components on Matterport3D dataset. ↓ indicates that lower is better; ↑ indicates that higher is better.

Modules	Error Metric ↓				Accuracy Metric ↑		
	**MRE**	**MAE**	**RMSE**	**RMSE log**	δ<1.25	$\delta <1.25^{2}$	$\delta <1.25^{3}$
Baseline	0.1151	0.2996	0.5039	0.0750	0.8712	0.9565	0.9815
Baseline + GSUM	0.1018	0.2681	0.4731	0.0732	0.8978	0.9641	0.9843
Baseline + GSUM + ACDCs	0.0981	0.2658	0.4709	0.0664	0.9027	0.9647	0.9848
Baseline + GSUM + AADCs	0.0958	0.2310	0.4661	0.0649	0.9068	0.9652	0.9856

**Table 3 sensors-23-09218-t003:** Performance comparison.

Approaches	Parameters	GPU mem.	GFLOPs	FPS
BiFuse [7]	253.1 M	4003 M	682.86	1
UniFuse [8]	30.26 M	1221 M	62.58	33
BiFuse++ [9]	53.19 M	1907 M	87.42	28
ACDNet [16]	87.0 M	2378 M	194.54	11
Ours	38.88 M	1739 M	54.27	37

## Data Availability

Data are contained within the article.

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
