# Peer review of "E2LNet: An Efficient and Effective Lightweight Network for Panoramic Depth Estimation"

_sensors, 2023, doi:10.3390/s23229218_

Round 1

Reviewer 1 Report

Comments and Suggestions for Authors

In the paper a method for panoramic depth estimation is presented. The problem being solved is well described, however some details of the method description must be improved. First of all, a better, theoretical motivation for the proposed architecture should be provided. In particular for AADC and GSUM modules. Secondly, the description of those modules must be improved. The figures and text are not consistent with each other. For example:

- It is not clear what are R^1,..., R^4 in Figure 2.
- In equation (1) there is F_i^k but I cannot find it in Figure 2.
- In Figure 1b there  is addition of one input?
- In Figure 3 there is F_i but in the text only F_4 is used (which is logical as GSUM works only on F_4).
- What is C in a circle in Figure 3?
- In bottom part of Figure 3 there is f_i but in the top part there is only f.

These are only examples of mentioned inconsistencies.

The other serious issue is connected with methodology of experiments. Authors do not provide information about dataset splits. There is usually train, validation and test set. In the text they mention, however, only about train and validation sets. Are the results in tables for calculated for validation or test set? Test set should be used there. Validation set is used only to prevent overfitting (hyperparmeter choice, early stopping, etc.) and should not be used for final assessment. Authors mention also that they use augmentation to prevent overfitting but anyway validation set should be used to check if it works.

Other comments:

- What was the resolution of processed images. 512x1024 or 256x512? It is not clear.
- In Figure 4 Accuracy was presented for which t?
- All figures have sub-figures (a), (b), etc., whereas in Figure 3 there is top and bottom part.
- English should be carefully checked e. g.: "panoramic depth estimation faces (...) distorion-aware", "pixel in an image through analysis", "a efficient", etc.
- There should be a space before cited articles ("something [1]" and not "something[1]").
- There should be no space in phrases like: "real -world", "multi-scale".
- Captions of Figures 5, 6 and 7 are the same. In general it is good to provide meaningful captions allowing to understand figure content. In those three cases authors could discuss what artifacts can be seen for presented methods.
- What are "Paras." in Table 3?
- Why in Table 2 two last modules are the same with different results?

Comments on the Quality of English Language

My comments are contained in general comments and suggestions.

Reviewer 2 Report

Comments and Suggestions for Authors

The authors propose a new framework for panoramic depth estimation. They added an attention mechanism and global scene understanding module into their proposed framework, which helps to perceive distortion and integrate global context information effectively. They claim that the proposed method achieves competitive performance comparable to existing techniques in both quantitative and qualitative evaluations. The topic seems interesting, and the author presents their findings well. I suggest some minor corrections based on the following points.

·   How effectively does the proposed adaptive attention dilated convolution module address the challenges associated with panoramic distortion, and how does it compare with traditional methods used to mitigate distortion in panoramic imaging?

·    In what specific ways does the global scene understanding module enhance the feature extraction process, and how does integrating global context information contribute to the accuracy of panoramic depth estimation in various real-world scenarios?

·  There are several typos that need to be fixed carefully. For example, “… contain 360°FoV …”, “ …  optimizer [55]  …”, “... PanoSUNCG [52].”, "... real-world datasets , Matterport3D ..." etc.

·   It is a good investigation to see the average recognition time and testing/recognition time for the proposed method and existing methods.

Comments on the Quality of English Language

Minor editing of English language required
